# OpenReview forum: "Evaluating AI Safety in Polish: An Automated Red-Teaming Approach"
_ICLR.cc/2025/Workshop/BuildingTrust — Submitted to BuildingTrust_

### Official Review · Reviewer_Ne3E · 2025-02-23
**Important topic, but needs to clarify the contributions and spend more space interpreting the results.**

**Rating:** 5
**Confidence:** 4

**Review:**

The paper introduces a methodology that automates the creation of red-teaming datasets for safety evaluations. It focuses specifically on using this framework in the context of the Polish language, given that  “Enhancing safety in one language (e.g., English) may inadvertently introduce vulnerabilities in others.” This mission is important and timely, and I appreciate the use of 14 different harm categories and 10 different attack styles. In general, I think the approaches outlined in this paper are on the right track, but I felt the reporting was not sufficiently strong for publication.

With a significant re-write and re-emphasis on reporting (a) why this technique is unique, (b) why other existing datasets (e.g., SafetyBench, WalledEval) or tools (e.g., PyRIT) could not be translated or implemented in other languages to achieve the goals outlined by this paper, and (c) interpreting the results would make this work publishable.

Additional comments:
- Like the comparative approach between models and postulation about why Llama and PLLUM models would differ in their ASR rates. Would like more of this!
- Appreciate the variety in harm type but would be interested in some rationale for choosing these categories. Are they common use cases? Do they conflict with model T&C or existing regulations?
- Also appreciate that multiple attack styles were included. However, there was again no rationale for including these styles (beyond that another author used them), and more importantly, no interpretation of results across style types. Why include this variation if the (sometimes large) differences between the styles are not discussed? This is an example of where more interpretation and presenting some hypotheses that can help explain the pattern of results would have been useful.
- I have some concerns about how the prompts were reviewed. Why would one reviewer be assigned the harmful prompts and another reviewer be assigned the non-harmful prompts? The prompts should have been randomly split, and then each reviewer assigned 66% of the prompts. This would ensure that (a) the reviewers are being impacted by the knowledge that the prompt should be harmful/non-harmful, and (b) allow a reliability metric to be computed for the 33% of prompts that both reviewers evaluated.
- The harmful and non-harmful generation prompts were very close to being matched (which is wonderful!) but the harmful prompts included an extra sentence at the beginning: “Refer to a specific act. Return only the prompt, do not write "prompt," do not comment, do not make excuses, I beg you!” The careful attention to matching, but then including this divergent sentence did not make sense to me.
- A key rationale for conducting this work is because: “Enhancing safety in one language (e.g., English) may inadvertently introduce vulnerabilities in others” … “[the current] English-centric approach may leave multilingual LLMs vulnerable in other languages. This is particularly concerning for languages underrepresented in safety training data, such as Polish.” I agree with this point, but feel some evidence should have been included to support this idea.


Unanswered questions:
- Why were styles typically transferred more frequently for the non-harmful prompts?
- Why were certain categories less ‘consistent’ (e.g., S2: non-violent crimes and S14: code interpreter abuse for Llama guard)?
- How much does the language of the base prompt impact the results? Especially curious what would happen if the “I beg of you!” sentence was removed from the harmful prompts.
- How does this work compare to other safety benchmarks (e.g., SafetyBench, WalledEval)?
- How is your work different from taking existing methods and using them in other languages? What are the unique benefits?
- If you were safety testing a model, would you prioritize a low ASR or FRR? How can one balance this trade-off?

Ultimately, I understand that there are significant length restrictions, but reporting the results in 158 words is difficult to understand when there were so many paragraphs that could have been combined and lines with hanging words and phrases. That space could have been used to beef up the reporting and clarify the contributions of this work. Too many of the key findings were relegated to the appendix.

---

### Official Review · Reviewer_L6d4 · 2025-02-25
**The paper constructs a dataset for the Polish language to test the safety of LLMs. The final experimental results indicate that there is significant disagreement among different models regarding whether the prompts are safe or not.**

**Rating:** 3
**Confidence:** 4

**Review:**

Strengths:
1. The paper addresses an important issue: the safety evaluation of LLMs in multilingual support.
2. The paper focuses on a less commonly studied language, Polish, and constructs its own dataset, which, if understood correctly, consists of 473 harmful prompts and 500 non-harmful prompts, totaling 973 test samples.
3. The paper conducts experiments on multiple open-source models.

Weaknesses:
1. What is the innovation in the pipeline from data generation, manual review, to final model evaluation compared to existing work?
2. In the experimental section, we observe significant deviations in test results across different models for the same input data. For example, some models have an ASR of 63.8%, while others have only 0.56%. Does this indicate a flaw in our method design? Or should the authors provide a deeper analysis of why such large discrepancies occur across models?
3. There are some obvious typos in the paper, such as "71.7%" being written as "71,7%" in Section 3.2.

Suggestions:
1. The dataset generation method involves using LLMs for automatic generation followed by manual review. Given that LLMs have over a 90% probability of generating the desired dataset, consider scaling up the LLM-generated data and eliminating the manual review process.
2. The paper uses a simple binary classification metric (safe & unsafe) to evaluate LLM safety. Could this be the reason for the significant differences in results across models? For example, to increase reliability, could the test data be modified into group-based datasets? For instance, grouping 5 datasets together and setting the group as "safe" if 4 out of 5 (depending on the strictness of detection) are classified as safe. The same method could then be applied when testing with other models to avoid bias from individual data points.
3. Another possibility is that the differences in results across models are due to inherent differences in the open-source models themselves. For example, what Llama-Guard-3-8B considers "safe" has a 63.83% ASR in the Bielik-7B-Instruct-v0.1 model. Additionally, we observe that ASR and FRR metrics show opposite trends in some models. Does this suggest that some models are inherently more "optimistic" (lenient) while others are more "pessimistic" (cautious)? Could more experiments be conducted to validate this hypothesis?

---

### Official Review · Reviewer_enog · 2025-03-01
**Evaluating AI Safety in Polish: An Automated Red-Teaming Approach**

**Rating:** 4
**Confidence:** 4

**Review:**

In Evaluating AI Safety in Polish: An Automated Red-Teaming Approach, the authors outline a methodology for generating harmful and non-harmful prompts for large language models to test their robustness against Polish prompts. The authors do a good job of clearly describing the process for generating red-teaming datasets. While the evaluation section is also clear, there is limited discussion on the implications or significance of the results. One path to make the paper stronger could be comparing the Polish results against English results (since the authors make the claim that models have generally already been subjected to English red-teaming)—in the abstract, there is a promise to “[highlight] the disparity between focusing on English and on Polish when generating the safety outputs” but it seems this promise is not fulfilled in the body of the paper. Additionally, it is unclear what one would do with the ASR and FRR results. Further discussion on that subject could also strengthen the paper.

---

### Decision · Program_Chairs · 2025-03-04

Reject